# Improvement of a Mathematical Model to Predict CO_2_ Removal in Hollow Fiber Membrane Oxygenators

**DOI:** 10.3390/bioengineering9100568

**Published:** 2022-10-18

**Authors:** Katelin S. Omecinski, William J. Federspiel

**Affiliations:** 1McGowan Institute for Regenerative Medicine, University of Pittsburgh, Pittsburgh, PA 15260, USA; 2Department of Bioengineering, University of Pittsburgh, Pittsburgh, PA 15260, USA; 3Department of Chemical and Petroleum Engineering, University of Pittsburgh, Pittsburgh, PA 15260, USA; 4Department of Critical Care Medicine, University of Pittsburgh Medical Center, University of Pittsburgh, Pittsburgh, PA 15260, USA; 5Clinical and Translational Science Institute, University of Pittsburgh, Pittsburgh, PA 15260, USA

**Keywords:** hollow fiber membrane bundle, oxygenator, extracorporeal oxygenation, extracorporeal CO_2_ removal, Haldane effect, artificial lung

## Abstract

The use of extracorporeal oxygenation and CO_2_ removal has gained clinical validity and popularity in recent years. These systems are composed of a pump to drive blood flow through the circuit and a hollow fiber membrane bundle through which gas exchange is achieved. Mathematical modeling of device design is utilized by researchers to improve device hemocompatibility and efficiency. A previously published mathematical model to predict CO_2_ removal in hollow fiber membrane bundles was modified to include an empirical representation of the Haldane effect. The predictive capabilities of both models were compared to experimental data gathered from a fiber bundle of 7.9 cm in length and 4.4 cm in diameter. The CO_2_ removal rate predictions of the model including the Haldane effect reduced the percent error between experimental data and mathematical predictions by up to 16%. Improving the predictive capabilities of computational fluid dynamics for the design of hollow fiber membrane bundles reduces the monetary and manpower expenses involved in designing and testing such devices.

## 1. Introduction

Chronic lower respiratory disease remains the fourth largest cause of death in the United States [1]. Extracorporeal membrane oxygenation (ECMO) or extracorporeal CO_2_ removal (ECCO_2_R) therapy is used to bridge acute lung failure patients to recovery or chronic lung failure patients to transplant. As these therapies have become increasingly clinically accepted and utilized, in part due to the COVID-19 pandemic, research into developing more compact, efficient, and hemocompatible devices has grown. The gas exchanging circuit component of ECMO or ECCO_2_R therapy is a structure composed of microporous hollow fiber membranes (HFM) woven into sheets and folded into bundle structures (HFM bundle); blood flows through the bundle around the fibers while a sweep gas, usually pure O_2_, flows through the fiber lumens. The juxtaposition of pure oxygen gas and venous blood creates a concentration gradient, causing O_2_ to diffuse from the sweep gas across the membrane and into the blood, and CO_2_ to diffuse across the membrane from the blood to the sweep gas. The efficiency of HFM bundles can be refined by iteratively modifying bundle characteristics, performing in vitro testing, and comparing experimental results. This trial-and-error method is costly in terms of money, materials, and manpower. Computational fluid dynamics streamlines the process by providing numerical insight into the fluid dynamics of each prototype. The most promising prototype can then be chosen and tested rather than multiple prototypes being tested and compared on the benchtop. This reduces the total time and manpower required to select the best performing prototype therefore reducing the overall cost of the research and development stage of HFM bundle design.

Mockros and Leonard [2] developed one such CFD model for use with compact cross-flow tubular oxygenators made of polypropylene. The model that predicts oxygenation rates in blood based on a dimensionless mass transfer correlation between Sherwood, Schmidt, and Reynold’s numbers. The correlation includes two empirical constants that were determined experimentally in water. To relate the mass transfer correlation developed in water to blood, the differences in oxygen storage within the two fluids had to be considered. Water carries oxygen only in a dissolved form that follows Henry’s law. Blood carries oxygen dissolved in the plasma as well as bound to hemoglobin in the form of oxyhemoglobin. An effective diffusivity was included in the Schmidt number to account for the convection of oxyhemoglobin. The effect of oxyhemoglobin on the solubility of oxygen in the blood was accounted for using the slope of the oxygen-dissociation curve in the Sherwood and Schmidt numbers. This curve represents the change in total blood oxygen content which occurs with changing oxygen partial pressure [3].

Svitek and Federspiel [4] took this process one step further by creating a dimensionless mass transfer correlation for both oxygen and carbon dioxide that collapsed onto one curve. This was done by utilizing the oxygenation relationships defined by Mockros and Leonard and developing additional diffusivity and solubility relationships to account for the differences in CO_2_ storage between water and blood. Carbon dioxide is also only carried by water in a dissolved form that follows Henry’s law while CO_2_ is stored in the blood in three ways: as bicarbonate (70%), bound to hemoglobin (23%), and dissolved in plasma (7%) [5]. Effective diffusivity in the Schmidt number accounts for the convection of carbon dioxide carried as bicarbonate and as protonated hemoglobin. Within the Sherwood number, facilitated diffusivity accounts for the diffusion of bicarbonate in addition to CO_2_ dissolved in the plasma. This facilitated diffusivity is not required for the oxygenation model as oxyhemoglobin only exists within an RBC and is therefore only carried by convection. The effect of protonated hemoglobin and bicarbonate on the solubility of CO_2_ into the blood was accounted for using the slope of the CO_2_ dissociation curve in both the Sherwood and Schmidt numbers. This curve represents the change in total blood carbon dioxide content which occurs with changing carbon dioxide partial pressure.

The description of the solubility of O_2_ and CO_2_ by empirical correlations for the oxygen- and CO_2_-dissociation curves are limitations of their respective models. In physiological gas exchange, several factors cause shifts in each respective dissociation curve. For the oxygen-dissociation curve, a decrease in pH or an increase in PCO_2_, blood temperature, and/or 2,3 DPG causes the curve to shift to the right [6]. The total blood hemoglobin concentration, Hb, also adjusts the location of the curve. The interaction between PCO_2_ and the oxygen-dissociation curve is known as the Bohr effect [7]. For the CO_2_-dissociation curve an increase in hemoglobin saturation, pH, Hb, and blood temperature results in a rightward shift. The interaction between oxygen concentration and the CO_2_ dissociation curve is known as the Haldane effect [8]. The Bohr and Haldane effects are not accounted for in the previous mathematical models as the dissociation curves used in the solubility calculations are only represented as univariate functions of the gaseous partial pressure. The Haldane effect is quantitatively more important to the transportation of CO_2_ than the Bohr effect is in promoting the transport of O_2_ [5]. Unbound hemoglobin is more basic than oxyhemoglobin and has a higher buffering capacity for hydrogen ions [9]. This results in increased removal of hydrogen ions from the RBC cytosol and favors the conversion of carbonic acid into protons and bicarbonate. Blood therefore has a larger overall storage capacity for CO_2_ compared to O_2_. As hemoglobin becomes saturated with O_2_ in the lungs, the abundant and rapid dissociation of protonated hemoglobin and carbaminohemoglobin results in a relatively large partial pressure gradient to drive molecular exchange. The reverse reaction is not as rapid, as H^+^ is a weaker acid than oxyhemoglobin, and the smaller reserve of O_2_ molecules in blood results in a smaller partial pressure gradient to drive mass transport. In this study we included the Haldane effect in the modeling of CO_2_ transfer and demonstrated a significant improvement in the ability to predict CO_2_ transfer in polymethylpentene HFM bundles.

## 2. Materials and Methods

The following analyses applies to a cylindrical axial fiber bundle with blood flow down the axis of the cylinder. A similar analysis can be done for annular fiber bundles with radial blood flow [4].

### 2.1. Steady State Mass Balance of Carbon Dioxide

The steady state mass balance for CO_2_ along the length of the bundle is:(1)$$Q_{b}\frac{dC_{CO_{2}}}{dz}=-A\left(a_{v}\right)k_{CO_{2}}\left(P_{CO_{2},b}-\bar{P_{CO_{2},\;\;g}}\right),$$
where $Q_{b}\;$ represents the blood flow rate, $C_{CO_{2}}$ represents the total concentration of CO_2_ in blood, A is the area of the bundle perpendicular to blood flow (i.e., frontal area), $a_{v}\;$ is the surface area to volume ratio of the bundle, $k_{CO_{2}}$ is the mass transport coefficient of CO_2_,$\;P_{CO_{2},b}$ is the partial pressure of CO_2_ in the blood, and $P_{CO_{2},\;g}$ is the partial pressure of CO_2_ in the sweep gas. $P_{CO_{2},\;g}$ is typically low compared to $\;P_{CO_{2},b}$, therefore the average of $P_{CO_{2},g}$ between gas flow inlet and outlet can be used [4].

The total concentration of CO_2_ in the blood is represented as a mathematical fit of the CO_2_ dissociation curve in the form [10]:(2)$$C_{CO_{2}}=qP_{CO_{2},b}^{t},$$
where *q* and *t* are regression parameters dependent on the oxygen content of the blood. The regression parameters are derived below under the section subheading Incorporating the Haldane effect.

Substituting Equation (2) into Equation (1) results in an equation of the form:(3)$$Q_{b}qt\left(P_{CO_{2},b}^{t-1}\right)\frac{dP_{CO_{2,b}}}{dz}=-A\left(a_{v}\right)k_{CO_{2}}\left(P_{CO_{2},b}-\bar{P_{CO_{2},g}}\right),$$

The mass transport coefficient, $k_{CO_{2}}$, is a constant that relates mass transfer rate, mass transfer area, and the difference in a partial pressure gradient that drives the movement of CO_2_ from the sweep gas to the blood. The mass transport coefficient of CO_2_ in the blood can be determined from an analogous heat transfer correlation for flow perpendicular to a bundle of tubes in the form [11]:(4)$$Sh=aRe^{b}Sc^{\frac{1}{3}},$$

The Sherwood number, Sh, relates the ratio of convective mass transfer to the rate of diffusive mass transport [12]. The Reynolds number, Re, is a ratio of inertial to viscous forces, and the Schmidt number, Sc, is the ratio of momentum to mass diffusivity [12]. The coefficients a and b are dependent on the geometry of the HFM bundle [4] and can be found in Table 1.

The Reynold’s number describing flow conditions of a fluid within a packed bed takes the general form [11]:(5)$$Re=\frac{V_{0}}{\gamma \psi a},$$
where $V_{0}$ is the superficial velocity through the HFM bundle and γ is the fluid viscosity. Superficial velocity is a hypothetical fluid flow that is calculated by dividing the volumetric flow rate of fluid through the bundle by the cross-sectional area of the HFM bundle. The characteristic length, 1/ψa, considers a correction factor for the geometry of the packing the bed, ψ=0.91, and the surface area of the fibers per unit volume of the bundle:(6)$$a=\frac{6\left(1-E\right)}{d_{p}},$$
where E is the bundle porosity and $d_{p}$ is the particle diameter [11]. For cylindrical particles, the HFM fibers, the particle diameter is expressed as:(7)$$d_{p}=0.567\sqrt{A_{f}},$$
where $A_{f}$ is the total surface area of the gas exchanging portion of the hollow fiber membranes [11].

The Sherwood number describing the flux of a gas into a fluid takes the general form:(8)$$Sh=\frac{kL}{\alpha D},$$
where *k* is the mass transport coefficient of the gaseous species, *L* is the characteristic length of the system, α is the solubility of the gas in the fluid, and D is the diffusivity of the gas into the fluid. For the flux of CO_2_ into blood, the mass transport coefficient is unknown and equal to $k_{CO_{2}}$, the characteristic length is the outer diameter of a single fiber, $d_{f}$, and the solubility of CO_2_ into blood is known and represented as $\alpha_{CO_{2}}$. The diffusivity, D,  must consider the diffusion of CO_2_ dissolved in the plasma and the diffusion of CO_2_ stored as bicarbonate. This value will be referred to as the facilitated diffusivity and is represented mathematically by [4,19]:(9)$$D_{f}=D_{CO_{2}}+\frac{D_{HCO_{3}}}{\alpha_{CO_{2}}}\frac{\delta C_{HCO_{3}}}{\delta P_{CO_{2},b}},$$
where $D_{CO_{2}}$ is the diffusivity of CO_2_ in blood, $D_{HCO_{3}}$ is the diffusivity of bicarbonate in blood, $\alpha_{CO_{2}}$ is the solubility of CO_2_ in blood, and $dC_{HCO_{3}}/dP_{CO_{2,b}}$ is the change in bicarbonate ion concentration with respect to partial pressure of CO_2_ in the blood. $dC_{HCO_{3}}/dP_{CO_{2,b}}$ is the slope of the CO_2_ dissociation curve, Equation (2), as the majority of carbon dioxide in the blood is stored as bicarbonate [4].

The Schmidt number takes the dimensionless form:(10)$$Sc=\frac{\nu_{b}}{D},$$
where $\nu_{b}$ is the kinematic viscosity of blood. The diffusivity, D, must account for the convection of CO_2_ stored as carbaminohemoglobin and bicarbonate. This value will be referred to as the effective diffusivity, $D_{eff,\;CO_{2}},$ and is represented mathematically by [4]:(11)$$D_{eff,CO_{2}}=\frac{D_{f}}{1+\frac{1}{\alpha_{CO_{2}}}\frac{\delta C_{HCO_{3}}}{\delta P_{CO_{2},b}}},$$

Substituting Equation (5) through (11) into the dimensionless correlation of Equation (4) and rearranging to solve for the mass transport coefficient results in the mathematical equation:(12)$$k_{CO_{2}}=\frac{\alpha_{CO_{2}}D_{f}aRe^{b}Sc^{1/3}}{d_{f}},$$

Blood oxygen saturation is solved for using a steady state mass balance on the HFM bundle with oxygen as the species of interest.

### 2.2. Steady State Mass Balance of Oxygen

The steady state mass balance for O_2_ along the length of the bundle is:(13)$$Q_{b}\frac{dC_{O_{2}}}{dz}=A\left(a_{v}\right)k_{O_{2}}\left(P_{O_{2},b}-\bar{P_{O_{2},\;\;g}}\right),$$
where $C_{O_{2}}$ represents the total concentration of O_2_ in blood, $k_{O_{2}}$ is the mass transport coefficient of O_2_, $P_{O_{2},b}$ is the partial pressure of oxygen in the blood, and $P_{O_{2},\;g}$ is the partial pressure of O_2_ in the sweep gas. $P_{O_{2},\;g}$ is ideally high compared to $\;P_{O_{2},b}$, therefore the average of $P_{O_{2},g}$ can be used [4].

The total concentration of O_2_ in blood is a combination of oxygen dissolved in the plasma and bound to hemoglobin. This can mathematically be represented as:(14)$$C_{O_{2}}=\alpha_{O_{2}}P_{O_{2},b}+C_{T}HbSO_{2},$$
where $\alpha_{O_{2}}\;$ is the solubility of O_2_ in blood, $C_{T}$ is the oxygen binding capacity of hemoglobin, Hb is the total hemoglobin level in the blood, and $SO_{2}$ is the percent of hemoglobin present in the form of oxyhemoglobin [14,17]. Substituting the derivative of Equation (14) into Equation (13) gives:(15)$$Q_{b}\left(\alpha_{O_{2}}+C_{T}Hb\frac{dSO_{2}}{dP_{O_{2},b}}\right)\frac{dP_{O_{2},b}}{dz}=A\left(a_{v}\right)k_{O_{2}}\left(P_{O_{2},b}-\bar{P_{O_{2},g}}\right),$$

$SO_{2}$ is a function of the partial pressure of oxygen in blood, approximated well by the Hill equation [15,20,21]:(16)$$SO_{2}=\frac{{\left(\frac{P_{O_{2},b}}{P_{50}}\right)}^{n}}{1+{\left(\frac{P_{O_{2},b}}{P_{50}}\right)}^{n}},$$
where n  and $P_{50}$ are constants dependent on the age and species of animal blood being tested and can be found in Table 1 [2].

Equation (4) can also be used to derive the mass transport coefficient of O_2_, however the appropriate values for oxygen must be substituted into the general dimensionless values of the Reynolds, Sherwood, and Schmidt numbers [4]. The Reynolds number, Equation (5), applies for both the CO_2_ and O_2_ mass balance as it is not dependent on any gaseous species-specific values. For the flux of O_2_ into blood, the mass transport coefficient in the Sherwood number is unknown and equal to $k_{O_{2}}$, the characteristic length remains as $d_{f}$, and the solubility of O_2_ in blood is known and represented as $\alpha_{O_{2}}$. For oxygenation there is no facilitated diffusivity as the oxyhemoglobin is contained within the red blood cell and is transported only by convection. Therefore, the diffusivity, D, is simply the diffusivity of oxygen in blood, $D_{O_{2}}.$ For the Schmidt number, the viscosity remains the same, $\nu_{b}$, and an effective diffusivity must also be defined as the convection of the oxygen bound to hemoglobin must be considered [4]. This is done by using the slope of the Hill equation, Equation (16), to approximate the slope of the oxyhemoglobin dissociation curve:(17)$$D_{eff,O_{2}}=\frac{D_{O_{2}}}{1+\frac{C_{T}}{\alpha_{O_{2}}}\frac{\delta C_{O_{2}}}{\delta P_{O_{2}}}},$$
$\delta C_{O_{2}}/\delta P_{O_{2}}$ is equivalent to the slope of Equation (14). Using Equation (5) through (8), (10), (14), (16), and (17) the mass transport coefficient, $k_{O_{2}}$, becomes [4]:(18)$$k_{O_{2}}=\frac{\alpha_{O_{2}}D_{O_{2}}aRe^{b}Sc^{1/3}}{d_{f}},$$

### 2.3. Incorporating the Haldane Effect

As previously stated, the CO_2_ dissociation curve can be represented by Equation (2). This equation is derived from a linear fit of any whole blood CO_2_ dissociation curve when plotted on logarithmic coordinates. The mathematical model that does not include the Haldane effect assumed a constant *q* and *t* value to define the CO_2_ dissociation curve throughout the entirety of the bundle. While this assumption greatly simplifies the mathematical calculations made within the model, it does not accurately reflect the compensatory mechanisms blood uses to achieve efficient CO_2_ removal. It is within this section that an iteratively updating CO_2_ dissociation curve will be included in the model to incorporate the Haldane effect.

In 1924 Peters et al. demonstrated that between the CO_2_ partial pressure values of 30 mmHg and 60 mmHg the slope of the CO_2_ dissociation curve is linearly related to the oxygen capacity of the blood being tested [10,22]:(19)$$C_{CO_{2}}\left(P_{CO_{2}}=60\right)-C_{CO_{2}}\left(P_{CO_{2}}=30\right)=0.334\left(O_{2}cap\right)+6.3,$$

This can then be related to hemoglobin concentration as it is known that 1.36 mL of oxygen combines with 1 g of hemoglobin [23]:(20)$$C_{CO_{2}}\left(P_{CO_{2}}=60\right)-C_{CO_{2}}\left(P_{CO_{2}}=30\right)=0.4542\left(Hb\right)+6.3,$$

Using Equation (2) in the definition of the slope of a linear function and substituting it for the left hand side of Equation (20) results in the mathematical equality [10]:(21)$$C_{CO_{2}}{\left(\frac{60}{P_{CO_{2,b}}}\right)}^{t}-C_{CO_{2}}{\left(\frac{30}{P_{CO_{2},b}}\right)}^{t}=0.4542\left(Hb\right)+6.3,$$
which can then be solved numerically if an ordered pair of ($P_{CO_{2},b},C_{CO_{2}})$ is known.

The initial value condition of $P_{CO_{2},b}$ is obtained from a blood gas analyzer or set of target venous conditions. In this model, pH is considered to be constant as the percent increase from typical venous to arterial pH is 0.5% [24]. An empirical mathematical equation relating these values was established by Visser in 1961 and further modified by McHardy in 1967 based on a nomogram of experimental data gathered by Van Slyke and Sendroy in 1928 [25].
(22)$$C_{CO_{2}}=2.226\;HCO_{3}^{-}\left[1-\frac{0.02924\left(Hb\right)}{\left(2.244-0.422SO_{2}\right)\left(8.74-pH\right)}\right],$$
where $HCO_{3}^{-}$ is the amount of carbon dioxide present in the blood in the bicarbonate form and $SO_{2}$ is a function of $P_{O_{2},b}$ according to Equation (16). Bicarbonate serves as a buffer to the acidic presence of CO_2_ in the plasma to maintain a physiologically safe blood pH. This acid-base homeostatic mechanism can be represented by the Henderson-Hasselbach equation in the form [26]:(23)$$HCO_{3}^{-}=0.0301\;P_{CO_{2},b}\left(1+10^{pH-6.10}\right),$$

Now with a single known value of $P_{CO_{2},b},$ $C_{CO_{2}}$ can be calculated and Equation (21) can be solved for the value of *t*. Equation (3) can be rewritten as:(24)$$q=\frac{C_{CO_{2}}}{P_{CO_{2},b}^{t}}$$
and *q* can be solved for with the now-known values of $C_{CO_{2}},\;P_{CO_{2},b},\;$ and *t*.

With Equations (19)–(24), the values of *q* and *t* are functions of $P_{CO_{2},b}\;$ and $S_{O_{2}},\;$ and change accordingly as blood traverses through the HFM bundle. With the inclusion of $S_{O_{2}}$ in the mass balance for carbon dioxide, Equations (1) and (13) become coupled first-order differential equations that can be solved numerically. The initial values of $S_{O_{2}}$ and $P_{CO_{2},b}$ are directly measured from the blood entering the ECCO_2_R device or a set of targeted venous conditions. The physical constants for O_2_ and CO_2_ are listed in Table 1 [4,13,14,15,16,18]. The above system of first-order differential equations was solved in MATLAB R2022A (MathWorks, Natick, MA, USA) using the Runge–Kutta fourth-order method [27].

## 3. Results

Experimental data from May et al. [18] are used as the experimental values in calculations of percent error. The range of blood flow rates tested in this publication are relative for ECCO_2_R where CO_2_ removal is the goal of therapy. The characteristics of the HFM bundle tested in that publication are listed in Table 2 [28,29,30]. Furthermore, presented is the predicted CO_2_ removal and oxygenation rates of the experimental HFM bundle mathematically calculated by the previous iteration of this model, where the Haldane effect is not accounted for.

The previous iteration of the mathematical model, without the Haldane effect, produced a predicted CO_2_ removal rate within 16% error at a blood flow rate of 240 mL min^−1^, 18% error at a blood flow rate of 500 mL min^−1^, and 30% error at a blood flow rate of 753 mL min^−1^. (Figure 1) The proposed mathematical model, including the Haldane effect, predicted a CO_2_ removal rate within 15%, 4%, and 14% error respectively.

The inclusion of the Haldane effect into the mathematical model melded the CO_2_ removal and oxygenation models as the mass balances became a paired set of first-order differential equations. However, only the CO_2_ model underwent modifications to enable the output of the oxygenation model, O_2_ content of the blood, to become an input variable for the CO_2_ mass balance. The oxygenation model was therefore unaffected by the change and both the code that incorporates the Haldane effect and the one that does not produce the same values of oxygenation at the same blood flow rates (Figure 2) [18]. At 240 mL min^−1^ blood flow rate the models predicted an oxygenation rate within 2% error, at 500 mL min^−1^ the prediction was within 10% error, and at 740 mL min^−1^ the prediction was within 2% error.

## 4. Discussion

With the acceptance and growing clinical use of ECMO and ECCO_2_R the need for more compact, efficient, and user-friendly systems has become a focus for medical product development. The performance of competing oxygenator designs can be quantitatively compared after manufacturing and benchtop testing. Researchers have turned to the use of computational fluid dynamics to evaluate oxygenator designs as this trial-and-error method requires a great deal of resources and manpower. The modifications proposed in this publication were intended to address a limitation of previously published oxygenation and carbon dioxide removal mathematical models. The Haldane effect was mathematically incorporated into these models via an empirical fit of the CO_2_ dissociation curve. This coupled the previously univariate models into a single multivariate model. The inclusion of the Haldane effect increased the accuracy of the CO_2_ removal predictions made by the model when compared to the same mathematical model with a static CO_2_ dissociation curve, i.e., without the Haldane effect. The improvement in the CO_2_ removal predictions is associated with the buffering capacity of single red blood cells.

The constant curve used by the model that does not account for the Haldane effect assumes blood is at an oxygen saturation of 100% and a Hb = 15 g (dL blood)^−1^. Testing conditions for oxygenators, however, typically dictate venous oxygen saturation to be 65% and blood hemoglobin to be 12 g (dL blood)^−1^ [13]. Oxyhemoglobin is a stronger acid than both unbound hemoglobin and protonated hemoglobin. As hemoglobin becomes oxygenated, protonated hemoglobin is forced to dissociate into unbound hemoglobin and a proton. Carbaminohemoglobin is also forced to dissociate, displacing additional intraerythrocytic CO_2_ into the plasma. Therefore, at the same total concentration of CO_2_ in the blood, a system with a greater oxygen saturation will store a greater percentage of CO_2_ within the plasma rather than within the red blood cell, i.e., the Haldane effect. This creates a larger partial pressure gradient to drive mass transfer between the blood and sweep gas, effectively increasing the rate of CO_2_ exchange achieved by the HFM bundle.

At the same pH, partial pressure of CO_2_, and SO_2_, a system with higher hemoglobin, 15 g (dL blood)^−1^ will have a smaller total concentration of CO_2_ than a system with lower hemoglobin, 12 g (dL blood)^−1^. As hemoglobin increases there is a resulting decrease in overall space in the plasma, an increase in the pH of blood, and a decrease in bicarbonate ions [21]. It has been shown, however, by May et al. [31] that increased hemoglobin results in increased CO_2_ removal rates in identical oxygenators at the same blood flow rate, inlet pH, inlet partial pressure of CO_2_, and inlet SO_2_. The increased CO_2_ removal but decreased overall CO_2_ content seems counterintuitive, however we are still seeing the Haldane effect at work. In a system with a Hb of 15 g (dL blood)^−1^ at a saturation of 65%, a total of 9.8 g (dL blood)^−1^ of Hb are present in the form of oxyhemoglobin. In a system with a Hb of 12 g (dL blood)^−1^ at a saturation of 65%, a total of 7.8 g (dL blood)^−1^ of Hb are present in the form of oxyhemoglobin. The increased presence of oxyhemoglobin forces a greater amount of protonated hemoglobin and carbaminohemoglobin to dissociate. Even though the overall content of CO_2_ in the 15 g (dL blood)^−1^ of Hb system is smaller than the alternative, a greater amount of it is stored within the plasma and there is a higher partial pressure gradient present to drive CO_2_ exchange mass transfer. In addition, the greater presence of oxyhemoglobin and the increased area available for the chloride shift results in an acceleration of the dissociation of the various forms of CO_2_ from hemoglobin. This is also compounded by a greater presence of Hb which provides an additional increase to the whole blood pH. There is therefore a greater change in total CO_2_ content per change in partial pressure of CO_2_ compared to a system with less Hb. Mathematically, the CO_2_ dissociation curve for a system with a higher Hb value will therefore have a greater slope at any one partial pressure of CO_2_ than a system with a smaller concentration of Hb.

In this manner utilizing the model without the Haldane effect results in a CO_2_ dissociation curve that is right-shifted and has a greater slope than is present in the physiological system. This ultimately results in an overestimation of the CO_2_ gradient that drives mass transfer and therefore the CO_2_ removal capabilities of a modeled HFM bundle. The empirical equations utilized to incorporate the Haldane effect are limited in accuracy to the range of conditions over which the physiologic data were gathered and the relationships were established. The assumption of the CO_2_ dissociation curve’s linearity on logarithmic co-ordinates (Equation (2)) has been verified between for CO_2_ partial pressure values of 20–80 mmHg by plotting human data gathered by Dittmer and Grebe and Henderson et al. [10]. Equation (19) was defined by Peters et al. in the 1920s, via a fit of experimental data, between CO_2_ partial pressure values of 30–60 mmHg [10,22]. As values stray from these ranges the predictive capabilities of the model may become less accurate. CO_2_ partial pressure values experienced during the collection of the presented in vitro data ranged from 9–45 mmHg. CO_2_ partial pressure values that were the farthest outside of the validated empirical range were experienced at a blood flow rate of 250 mL min^−1^. This correlates to the highest experienced percent error, 16%, between code predictions and experimental data. The code without the Haldane effect also experienced a 16% error at a blood flow rate of 250 mL min^−1^, but the highest experienced percent error was 30% at a blood flow rate of 753 mL min^−1^. The inclusion of the Haldane effect may therefore still provide more accurate predictions of CO_2_ removal, when used outside of the validated ranges, when compared to the code without the Haldane effect. Future work is needed to determine the extent to which the range can be exceeded without intolerably affecting results.

Employing computational fluid dynamics to evaluate hypothetical HFM bundle geometries is only as useful as the models are accurate. By incorporating a multivariate representation of the CO_2_ dissociation curve into a model that previously excluded it, our group improved the accuracy of predictions of HFM bundle CO_2_ removal capabilities. As the understanding of the mass transfer principles of hollow fiber membranes, HFM bundles, and blood continues to grow, so does the ability of computational fluid dynamics to reduce the monetary and manpower requirements of device design.

## Figures and Tables

**Figure 1 bioengineering-09-00568-f001:**
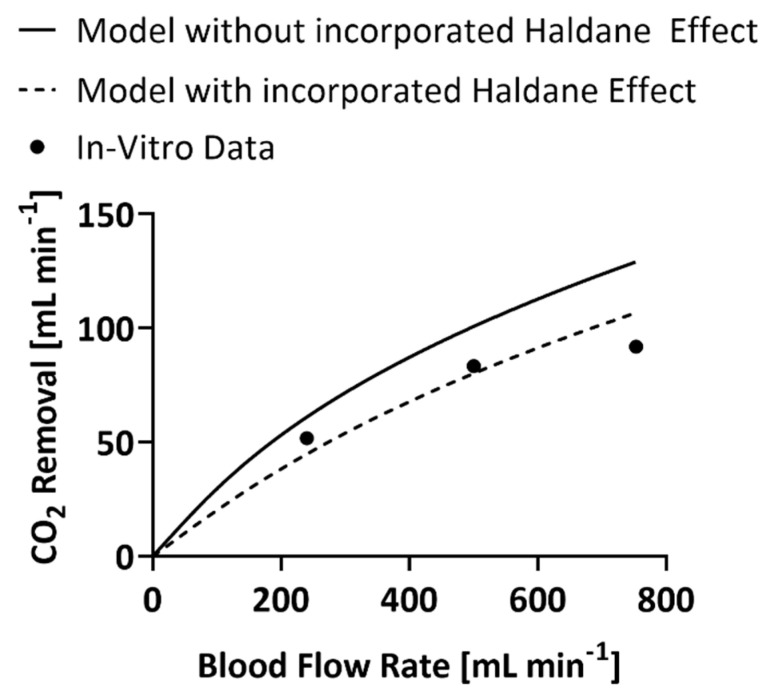
In vitro values of the CO_2_ removal rate of the experimental HFM bundle compared to predictions of CO_2_ removal by the model that includes the Haldane effect and the one that does not [18]. Error bars are included but are too small to be seen.

**Figure 2 bioengineering-09-00568-f002:**
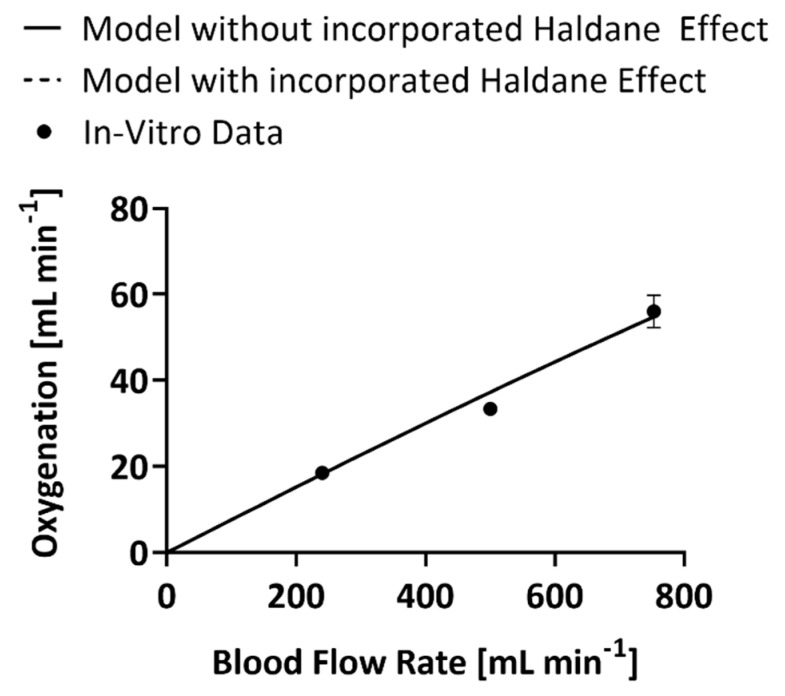
In vitro values of the oxygenation rate of the experimental HFM bundle compared to predictions of oxygenation by the model that accounts for the Haldane effect and the one that does not [18].

**Table 1 bioengineering-09-00568-t001:** Physical Constants for O_2_ and CO_2_.

Parameter	Description	Value
$P_{CO_{2},b}\left(z=0\right)$	Value of $P_{CO_{2}}$ in the blood entering the HFM bundle	45 mmHg [13]
$S_{O_{2},b}\left(z=0\right)$	Initial value of $S_{O_{2}}$ in the blood entering the HFM bundle	65% [13]
$P_{CO_{2},g}$	Average value of $P_{CO_{2}}\;$ in the sweep gas	4 mmHg [4]
$P_{O_{2},g}$	Average value of $P_{O_{2}}$ in the sweep gas	700 mmHg [4,14]
Hb	Hemoglobin of blood	12 g (dL blood)^−1^ [13]
pH	pH of blood entering the HFM bundle	7.4 [5]
$P_{50}$	$P_{O_{2}}$ at 50% Hb saturation for adult bovine blood	29 mmHg [15]
n	Hill parameter for adult bovine blood	2.85 [15]
γ	Kinematic viscosity of blood	0.023 cm^2^s^−1^ [4]
$\alpha_{CO_{2}}$	Solubility of CO_2_ in blood	6.62 × 10^−4^ (mL CO_2_) (mmHg) (mL blood)^−1^ [4]
$\alpha_{O_{2}}$	Solubility of O_2_ in blood	3 × 10^−2^ (mL O_2_) (mmHg) (mL blood)^−1^ [4,14]
$D_{CO_{2}}$	Diffusivity of CO_2_ in blood	7.39 × 10^−6^ cm^2^s^−1^ [4]
$D_{HCO_{3}^{-}}$	Diffusivity of bicarbonate in blood	4.62 × 10^−6^ cm^2^s^−1^ [4]
$D_{O_{2}}$	Diffusivity of O_2_ in blood	1.8 × 10^−5^ cm^2^s^−1^ [4,14]
$C_{T}$	Binding capacity of hemoglobin	1.34 mL O_2_ (g Hb)^−1^ [5,16,17]
$Q_{b}$	Blood flowrate	0–600 (mL blood) min^−1^ [18]
a	Measured coefficient for Equations (4), (12), and (18)	0.54 [4]
b	Measured coefficient for Equations (4), (12), and (18)	0.42 [4]

**Table 2 bioengineering-09-00568-t002:** Characteristics of ModELAS HFM bundle.

Parameter	Description	Value
A	Cross sectional area of ModELAS HFM bundle	16 cm^2^
$a_{v}$	Surface area to volume ratio of ModELAS HFM bundle	55 cm^−1^
$A_{f}$	Active fiber surface area of ModELAS HFM bundle	6700 cm^2^
df	Outer diameter of a single OXPLUS™, Membrana™ PMP fiber	0.038 cm
E	Porosity of ModELAS HFM bundle	0.48

## Data Availability

The original data were available from the corresponding author upon an appropriate request.

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
