# Peer review of "Improvement of a Mathematical Model to Predict CO2 Removal in Hollow Fiber Membrane Oxygenators"

_bioengineering, 2022, doi:10.3390/bioengineering9100568_

Round 1
Reviewer 1 Report
This paper describes an attempt to improve the mathematical modelling to predict CO2 removal by hollow fiber membrane oxygenators. This is an important medical technology topic, and the presented approach and results can contribute to this field. The paper is suggested to be accepted for publication after some revision on the basis of comments below.
COMMENTS
1.
In the Introduction, the authors summarize the general background of oxygen and CO2 solubility, dissociation and exchange, and the related models. However, these properties also depend on the applied hollow fiber materials and technologies applied for their production. The authors should discuss this matter in the Introduction as well.
2.
It is unclear where the minus sign comes from in Equation 3.
3.
The authors list the applied parameters in Table 1. However, no literature sources are provided in this Table. Although the source of these parameters are given later (line 248), the authors should give in details the literature references for all these parameters in the Table and/or in the caption as well.
4.
In the caption of Figure 1, the literature source should be provided for the data depicted in this Figure along with the model calculations.
5.
In the caption of Figure 1, the authors write that “Standard deviations are included”. However, neither the standard deviations are given in this Figure nor the error bars. These should be included in this Figure.
6.
In the Abstract and at the end of the Discussion, the authors claim that computational fluid dynamics reduce the monetary and man-power requirements of device design. However, no any specific argument or example is provided how such monetary and man-power reduction would operate in practice. The authors should specifically comment on this matter in this manuscript.
Reviewer 2 Report
The Manuscript (bioengineering-1941626) entitled "Improvement of a Mathematical Model to Predict CO2 Removal in Hollow Fiber Membrane Oxygenators" submitted to bioengineering focused on the improvement of prediction for Hollow Fiber Membrane Oxygenators. The obtained results are interesting but the minor revision is needed according to the following comments prior to publishing.
v Red highlight: wrong spelling or wrong Eq (refere to the attached file)
1. Hemocompatability= hemocompatibility
2. facilitated diffusivity or facilitated diffusion?
3. quantitively= quantitatively
4. Eq15 Hb is constant like Ct, why Hb was omitted?
Other comments:
The Eq.14 is wrong because it had 2 main parts and So2 is percent of hemoglobin present in the form of oxyhemoglobin besides it doesn’t mention any references therefor it should be revised in the correct form.
Line 79: Causes the curve to shift to the right= it was better that it was showed curve
Line 83: Shift to right or left?
Line 143: The references of constants should be added for Table 1.
